# Critical Evaluation of the Methods for the Characterization of the Degree of Sulfonation for Electron Beam Irradiated and Non-Irradiated Sulfonated Poly(ether ether ketone) Membranes

**DOI:** 10.3390/ma16186098

**Published:** 2023-09-06

**Authors:** Laura Dace Pakalniete, Elizabete Maskova, Rudolfs Janis Zabolockis, Liga Avotina, Einars Sprugis, Ingars Reinholds, Magdalena Rzepna, Guntars Vaivars, Elina Pajuste

**Affiliations:** 1Institute of Chemical Physics, University of Latvia, Jelgavas Iela 1, LV-1004 Riga, Latvia; laura_dace.pakalniete@lu.lv (L.D.P.); liga.avotina@lu.lv (L.A.); einars.sprugis@lu.lv (E.S.); guntars.vaivars@lu.lv (G.V.); elina.pajuste@lu.lv (E.P.); 2Faculty of Chemistry, University of Latvia, Jelgavas Iela 1, LV-1004 Riga, Latvia; ingars.reinholds@lu.lv; 3Institute of Solid State Physics, University of Latvia, Kengaraga Iela 8, LV-1063 Riga, Latvia; 4Baltic Scientific Instruments, Ganibu Dambis 26, LV-1005 Riga, Latvia; 5Institute of Food Safety, Animal Health and Environment “BIOR”, Lejupes Iela 3, LV-1076 Riga, Latvia; 6Centre for Radiation Research and Technology, Institute of Nuclear Chemistry and Technology, Dorodna 16, 03-195 Warsaw, Poland; m.rzepna@ichtj.waw.pl

**Keywords:** sulfonated poly(ether ether ketone), degree of sulfonation, electron beam irradiation

## Abstract

Sulfonated poly(ether ether ketone) (SPEEK) materials are promising candidates for replacing Nafion™ in applications such as proton exchange membrane (PEM) and direct methanol fuel cells. SPEEK membranes have several advantages such as low cost, thermal and radiation stability and controllable physicochemical and mechanical properties, which depend on the degree of sulfonation (DS). Commercial PEEK was homogenously sulfonated up to a DS of 60–90% and the membranes were prepared using a solvent casting method. Part of the samples were irradiated with a 10 MeV electron beam up to a 500 kGy dose to assess the ionizing radiation-induced effects. Both non-irradiated and irradiated membranes were characterized by Fourier Transformation infrared (FT-IR) spectroscopy, thermogravimetric analysis (TGA), proton nuclear magnetic resonance (^1^H-NMR) spectroscopy, electrochemical impedance analysis and, for the first time for non-irradiated membranes, by spectrophotometric analysis with Cr(III). The above-mentioned methods for application for DS assessment were compared. The aim of this study is to compare different methods used for the determination of the DS of SPEEK membranes before and after high-dose irradiation. It was observed that irradiated membranes presented a higher value of DS. The appearance of different new signals in ^1^H-NMR and FT-IR spectra of irradiated membranes indicated that the effects of radiation induced changes in the structure of SPEEK materials. The good correlation of Cr(III) absorption and SPEEK DS up to 80% indicates that the spectrophotometric method is a comparable tool for the characterization of SPEEK membranes.

## 1. Introduction

The need for sustainable and environmentally friendly energy cannot be ignored considering the rapid depletion of fossil fuel resources [1,2,3]. Hydrogen energy-based systems are one of the leading technologies that can provide the requirements for such types of energy sources. Fuel cells are discussed as the most promising technological approaches due to a number of advantages such as:Low operational costs.Reduced harmful emissions down to zero.Robust technology.Improvements of efficiency with the change of fuel cell materials [4].

The efficiency of fuel cells is higher than that of other energy generation systems due to the properties of the proton exchange membrane fuel cells (PEMFC)—high power density, great durability, low operating temperature and rapid response to changes in system conditions [5,6,7,8,9,10,11].

Fuel cells have found an application in nuclear facilities for separating and recovering the heavy and radioactive hydrogen isotope, tritium [12,13]. Electrolytically processing water allows it to be enriched with the radioactive hydrogen isotope, tritium, and in such conditions the PEM is exposed to ionizing radiation. As tritium is often a main fuel in nuclear fusion reactors, it is important for the PEM to have great radiation stability and be able to function properly under the influence of ionizing radiation [14].

SPEEK polymer membranes have exhibited good chemical stability in fuel cell tests [15,16] and vanadium redox flow batteries [16,17]. The degree of sulfonation (DS) of this PEM determines how the membranes conducts protons, as well as the mechanical and chemical stability of the material [18,19]. By increasing the DS, proton conductivity is also increased, and that enhances its performance in PEMFCs. However, an increase in DS promotes quicker membrane deterioration, both chemical and mechanical, which in turn decreases proton conductivity [20].

SPEEK polymers (Figure 1), the sulfonated poly(ether ether ketone) membranes, have been investigated as polymer electrolyte materials due to their relatively lower costs and advanced properties in contrast to other currently commercially available PEMs, such as Nafion™ [21,22,23]. As the performance of PEMs is highly dependent on the sulfonation degree, a reliable and fast method for the determination of DS is required [14,22,23,24].

Due to its good radiation stability, SPEEK can also be used in radiation environments, such as nuclear facilities, space applications and proton exchange membrane-based enrichment of the hydrogen radioactive isotope, tritium [25,26]. Insufficient research has been performed to determine the effects of ionizing radiation on the DS, as well as how electron beam radiation affects the structure of the membrane, which affects the determination of DS.

In this study, synthesis of SPEEK membranes with various degrees of sulfonation and the determination of the DS using various spectrometric and analytical methods was carried out for both non-irradiated and electron-beam-irradiated SPEEK membranes.

## 2. Materials and Methods

### 2.1. SPEEK Synthesis

PEEK in granular form was purchased from Sigma-Aldrich (St. Louis, MO, USA). PEEK pellets were dried in a vacuum oven at 100 °C overnight. A total of 10 g of the pellets was added slowly to 200 mL of concentrated sulfuric acid (95–97%) with heating and vigorous stirring. The time and temperature control were adjusted to reach the desired DS of SPEEK, which ranged between 60 and 90%. After completing the necessary period of sulfonation (see Table 1), the reaction was terminated by pouring the sulfonated polymer directly into ice-water. The polymer precipitate was filtered and washed several times with deionized water until its pH reached 7. The filtered polymer was then dried under a vacuum at 60 °C for one week until it reached a constant weight [27,28]. SPEEK membranes were produced by dissolving obtained highly sulfonated SPEEK materials in *N*,*N*-dimethylformamide (DMF) and pouring them into Petri dishes, followed by drying for 48 h at 80 °C. The SPEEK membranes were removed from the Petri dishes and used for carrying out further experiments.

### 2.2. SPEEK Irradiation

The dried cast membranes were irradiated via a 10 MeV electron beam at the Institute of Nuclear Chemistry and Technology (Warsaw, Poland) with a total absorbed dose of 500 kGy. Dosimetry was carried out using a graphite calorimeter according to [29].

### 2.3. Impedance Analysis

Impedance analysis of the membranes was performed in two electrode through-plane configuration (the electrode diameter was 1 cm). Multichannel potentiostat/galvanostat VMP3 (BioLogic, Memphis, TN, USA) was used, and the measuring parameters frequency range was 50 kHz to 1 Hz; 10 frequencies per decade; signal amplitude 10 mV. The resistance with precision ±1 Ohm was obtained from a Nyquist plot extrapolated to the high frequencies [30].

### 2.4. Thermogravimetry Analysis (TGA)

TGA measurements were performed using a TGA1/SF thermogravimetric instrument (MettlerToledo (Columbus, OH, USA). Samples of SPEEK membranes (10 mg weight) were placed in alumina crucibles and thermally treated under air flux (50 mL/min) from 25 to 600 °C with a heating rate of 10 °C/min. For each membrane, three samples were prepared and analyzed as described above. The average TGA curves of the weight loss versus temperature and the derivative (DTG, %/°C) were analyzed for each membrane.

TGA data of the sulfonate group decomposition were directly used to calculate DS for SPEEK membranes. The Equation (1) described in the literature [28] was used:(1)$$\mathrm{DS}=\frac{n\left({\mathrm{SO}}_{3}H\right)}{n\left(\mathrm{PEEK}\right)}=\frac{M\left(\mathrm{PEEK}\right)}{\left(\frac{m}{\Delta m}\right)M\left({\mathrm{SO}}_{3}H\right)}$$
where M(PEEK) and M(SO_3_H) are the molecular masses of PEEK monomer (288.7 g/mol) and sulfonic acid groups (81 g/mol), m is the mass of SPEEK at the beginning of the desulfonation, and Δm is the mass loss due to the desulfonation [31].

### 2.5. Fourier-Transform Infrared (FT-IR) Spectrometry and FT-IR-TGA

Bruker Vertex 70v (Billerica, MA, USA) vacuum infrared spectrometer equipped with an attenuated total reflection (ATR) diamond accessory was used in this study. The recording range was 400–4000 cm^−1^, spectral resolution ±2 cm^−1^, in a 2.95 hPa vacuum, at least 3 measurements per sample, 20 spectra per measurement, obtaining total of at least 60 spectra for each of the SPEEK membranes. Average absorbance values were obtained by analyzing 3 different pieces of each of the corresponding membranes. The average spectrum was calculated from the measured three replicate spectra. Data were collected using TRIOS Software v4.3.1 and FT-IR program OPUS by Bruker, analyzed within OriginPro v8.0 scientific graphing and data analysis software.

### 2.6. Nuclear Magnetic Resonance Spectroscopy

The degree of sulfonation was determined using a ^1^H-NMR spectrum acquired with a Bruker Fourier-300 spectrometer (Billerica, MA, USA). An amount of 5–10 mg of the membranes was dissolved in deuterated dimethyl sulfoxide (DMSO-*d*_6_) solution, and its spectrum acquired (Figure 2). The DS was calculated in the MestReNova program using the ratio of peak areas of the proton peaks near the keto-group of SPEEK to the ratio of the proton next to the -SO_3_H group. A modified version of the formula presented by Parnian et Al was used. 

### 2.7. Spectrophotometry with Cr(III) 

Metals such as Fe(III) and Cr(III) can form ionic bonds with sulfonic acid groups present in PEMs [32,33,34]. Therefore, a novel method was developed to quickly and inexpensively determine the degree of sulfonation of proton exchange membranes (in this study, SPEEK) via photometric analysis of Cr(III). 

Chromium(III) readily reacts with disodium ethylenediaminetetraacetic acid (EDTA) at temperatures ~373 K to form a brightly colored purple complex that can be used to determine the Cr(III) ion concentration in solutions using spectrophotometric analysis [35].

To determine the DS, the membranes (0.005–0.01 g) were submerged in chromium (III) nitrate solutions of known concentration and volume for 24 h and light absorption measurements were performed with a Jenway 6300 spectrophotometer using 540 nm (maximum of absorption for chromium (III) complexonate [35]). Before the initial measurement of the various membrane samples, a calibration with standard solutions of Cr(III) ions was carried out. The Cr(III) standards were prepared in concentrations 0.04; 0.08; 0.12; 0.16 and 0.20 g/L. Acetate buffer solution (pKa = 4.7) and 5% EDTA solution was added. The obtained solutions were heated up to 373.15 K to obtain a purple color and then diluted to a known volume using a volumetric flask. The light absorption of all standards and samples was measured in 1 cm plastic cuvettes. Each standard and sample was measured 3 times for 30 s and the values were recorded. 

The decrease in concentration from the standard solution in which the membranes were submerged was calculated from the calibration chart obtained from the standard solutions.

## 3. Results and Discussion

### 3.1. Impendence Analysis

Figure 3 presents a box plot of the calculated proton conductivity measurements for both unirradiated and irradiated SPEEK membranes. The box plot is constructed from five values: the minimum value, the 25% quartile, 50% (median), the 75% quartile, and the maximum value for each type of membrane characterized by impedance analysis. It can be seen from Figure 3 that conductivity of the non-irradiated membranes changes proportionally to the increase in the DS. An increase in acidic groups promotes hydrophilic interactions, resulting in increased water absorption due to hydrogen bond formation. The absorbed water forms even more pathways for protons. By increasing the number of sulfonate groups in the polymer and, by extension, the membrane, hydrophilicity is increased, increasing water absorption and facilitating proton transport [16,36,37]. In the case of the electron-beam-irradiated SPEEK membranes, a tendency for increased conductivity is presented, with the SPEEK membrane of DS 90% showing the highest proton transport capability. The increase in the proton conductivity appears to be much steeper and quicker for irradiated membranes than their non-irradiated counterparts, indicating a change in structure that facilitates the transport of protons.

### 3.2. Thermogravimetry Analysis (TGA)

The TGA curves of non-irradiated and irradiated SPEEK membranes are expressed in Figure 4 and Figure 5 and show three thermal transition stages for all membranes. The first mass loss can be attributed to water evaporation, the second one to the desulfonation reaction: 4SO_3_H → 4SO_2_ + 2H_2_O + O_2_ and the last one can be attributed to oxidative pyrolysis of the PEEK chain, forming H_2_O and CO_2_ upon decomposition [38]. It is likely that due to varied distribution of the sulfonic acid groups in the polymer, they decompose at slightly different temperatures. According to the TGA data, the DS slightly increased after irradiation (Figure 5). This might be related to the radiation-induced sulfonation from the unreacted acid in the membrane structure.

The DS was calculated using the TGA curves and Equation (1). The calculated DS can be seen in Table 2.

### 3.3. Fourier-Transform Infrared (FT-IR) Spectroscopy and FT-IR-TGA

FT-IR spectra of the non-irradiated SPEEK membranes with varying sulfonation degrees are summarized in Figure 6. The normalysed spectra are shown in Figure 7 and the spectra for irradiated SPEEK membranes are shown in Figure 8.

Interpretation of the bonds is based on information found in the literature [16]. The absorption axis is shifted for visualization (displayed in relative units). The most intense signal for SPEEK membranes with the sulfonation degree 60–90% is determined to be at 1158 cm^−1^. This signal is selected for the normalization of the spectra. After the normalization, some tendencies are observed—the changes in the intensities of some signals that are correlating with the sulfonation degree of the membranes (Figure 7). The signal values according to the sulfonation degree are summarized in Table 2. As a preview, for the signal at 1185 cm^−1^, the intensity in 60% sulfonated membrane is around 88% in comparison to maximum, while in the membrane with the sulfonation degree of 90%, the intensity of the particular signal is decreased to 81% in comparison to the maximum.

The bond vibrations observed at 1185 cm^−1^ in the FT-IR spectrum correspond to asymmetric and symmetrical O = S = O stretching vibrations. The sharp absorption peak at 1596 cm^−1^ is attributed to the aromatic C = C vibration [39]. It is shown in Figure 9 and Figure 10 that these two characteristic peaks show a linear decrease in absorption intensity when the degree of sulfonation is increased, with the R^2^ of the non-irradiated values for 1185 and 1596 cm^−1^ peaks being 0.90 and 0.93, and for irradiated values, 0.99 and 0.97, respectively.

It can be observed from the Figure 9 and Figure 10 that both irradiated and non-irradiated membranes show an R^2^ value that is equal to or higher than 0.90, indicating that there is a sufficient correlation for the characterization of the DS based on peak intensities.

The correlation between peak intensities and DS is much steeper for irradiated membranes, indicating that there has been a change in the membrane’s structure after irradiation, causing slightly differing peak intensities at these wavenumber values.

### 3.4. ^1^H-NMR

The close-up of the integrated ^1^H-NMR spectra can be seen in Figure 11 and Figure 12. The DS was determined by setting the integral of the H_1_, H_2_, H_3_ and H_4_ protons to four (shown in Figure 11), as their peak intensities are not affected by DS, and the peak integral of the H_5_ proton directly corresponds to the DS of the SPEEK membrane (Table 3) [40].

### 3.5. Spectrophotometry with Cr(III) 

It can be seen that a higher DS correlates with a lower leftover Cr(III) mass concentration compared to the original solution due to the reaction of Cr(III) ions with -SO_3_H groups. The pKa of SPEEK is reported as 1.58 for SPEEK with DS 85% in [41], meaning that above pH 1.58 SPEEK -SO_3_H groups will have already displaced the protons creating an ionic bond between the -SO_3_^−^ group and Cr(III) atoms. The obtained pH values for SPEEK samples with DS 60%, 70%, 80% and 90% were as follows: 2.63, 2.84, 1.94, 2.32. As Cr(III) ions carry a 3+ charge, three sulfonic acid groups are expected to be attached to one Cr atom. 

The calibration curve results can be seen in Figure 13. A decreasing linear correlation can be observed in Figure 14 and Figure 15 from DS 60% to 80%, indicating that the method is valid for the determination of the DS for SPEEK membranes with DS < 90%. Figure 16 illustrates one of the possible bond creation mechanisms for SPEEK and Cr(III).

One explanation for the deviation from the linear correlation with DS 90% could be that the polymer starts dissolving in the aqueous medium, inhibiting the reaction with Cr(III) ions.

### 3.6. Summary of the Results

There is a tendency for the proton conductivity of SPEEK membranes to increase with both DS and irradiation, indicating that there is a possibility that electron-beam-irradiated membranes undergo structural changes after irradiation that are significant enough to improve the conductivity of protons. 

Electron-beam-irradiated SPEEK membranes showed a different degree of sulfonation than non-irradiated SPEEK membranes with TGA, which indicates that the irradiated membranes might have either undergone electron-beam-induced crosslinking or radiation induced sulfur addition from the remaining acid.

The abovementioned correlation between the sulfonation degree and FT-IR spectra peak intensities can be considered for application of the FT-IR ATR method not only for the qualitative determination of the presence of functional groups in the SPEEK membranes, but also for quantification. In order to correctly determine the DS with FT-IR, a calibration graph using multiple reference membranes with known DS should be obtained, and then normalized based on the most intense peak, which was determined to be at 1158 cm^−1^ for this experiment series. The DS can be obtained by comparing the intensities of the normalized signals in the calibration graph. 

^1^H-NMR integration results showed a higher DS on average than TGA, and because of the manual integration aspect, determining the DS with ^1^H-NMR can lead to more inaccuracies. Figure 17 and Figure 18 both show an overview for non-irradiated and irradiated SPEEK membrane DS results by method.

A calibration curve was obtained with Cr(III) standard solutions and it was determined that this method can be used to determine the DS of SPEEK with values above 60% and up to 80%, as that is where the linear range ends.

The advantages and the limitations of the evaluated methods for the characterizadion of SPEEK DS are summarized in Table 4. 

## 4. Conclusions

In this work, various methods for determining the degree of sulfonation for non-irradiated SPEEK membranes were applied and compared. 

Direct measurement is possible with TGA and ^1^H-NMR methods; however, both require sophisticated equipment and are destructive. The FT-IR method requires calibration with the known DS samples; it also requires sophisticated equipment, but it is a comparably fast method and is non-destructive. 

Spectrophotometric determination of the DS using Cr(III) can be achieved for a DS that is lower than 90%, as the linear region was observed to end at 80% DS, but a calibration curve with known concentrations of Cr(III) in a solution should be obtained first.

It was observed that the DS values after irradiation in most cases were elevated by 8%, showing that electron beam radiation influences the structure of the membrane. However, for absorbed doses up to 500 kGy, SPEEK remains stable and is suitable for beta negative radiation environments. The biggest change was observed in SPEEK_4 with DS 90%, while the smallest was in SPEEK_1 with DS 60%, indicating that after irradiation, the higher the DS, the larger the change in the membrane’s structure, possibly due to higher sulfonic acid group counts providing more opportunities for irradiation-induced polymer structure change.

## Figures and Tables

**Figure 1 materials-16-06098-f001:**
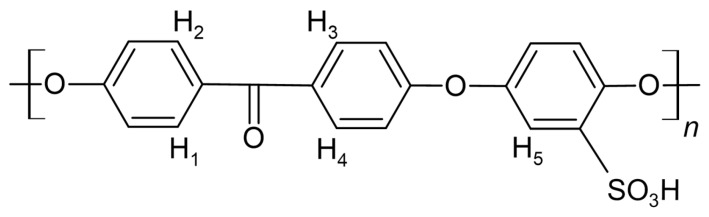
SPEEK monomer marked with protons used for integration.

**Figure 2 materials-16-06098-f002:**
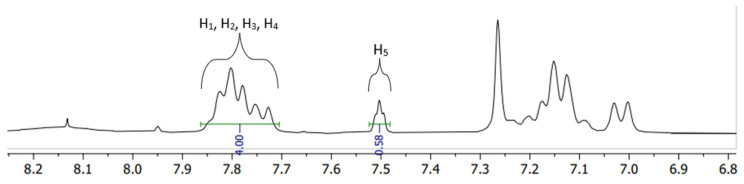
The integrated ^1^H-NMR spectra example.

**Figure 3 materials-16-06098-f003:**
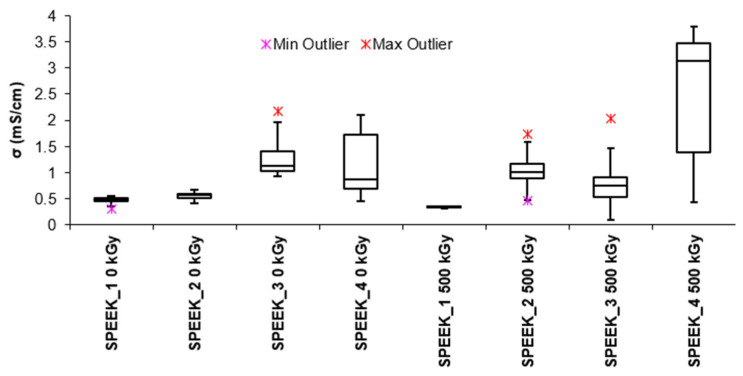
Conductivity of irradiated and non-irradiated SPEEK membranes depending on the DS of the SPEEK.

**Figure 4 materials-16-06098-f004:**
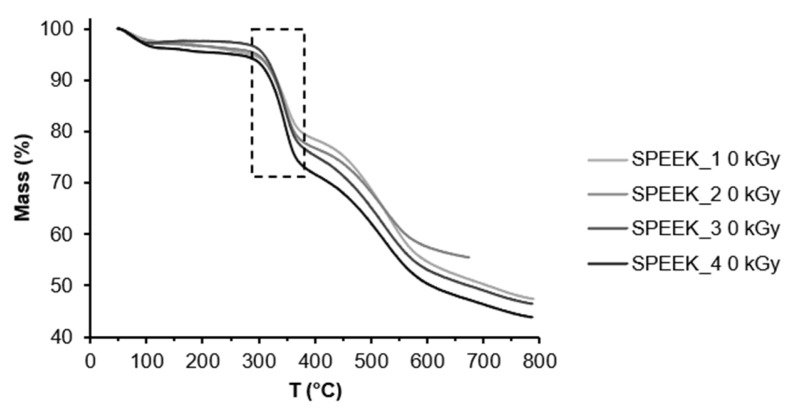
TGA curve for non-irradiated SPEEK samples with different DS.

**Figure 5 materials-16-06098-f005:**
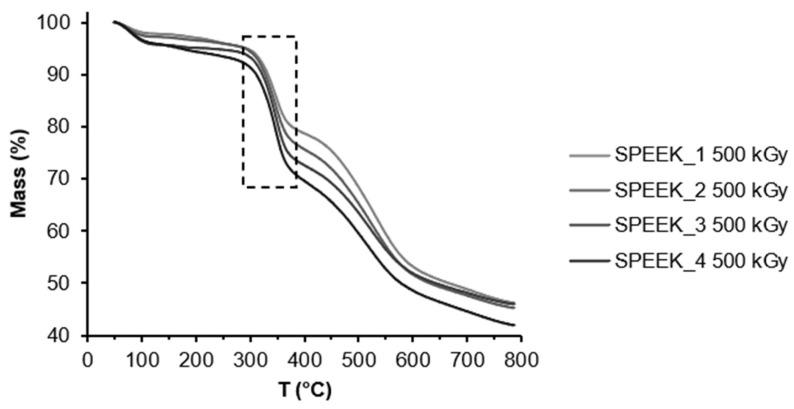
TGA curve for irradiated SPEEK samples with different DS.

**Figure 6 materials-16-06098-f006:**
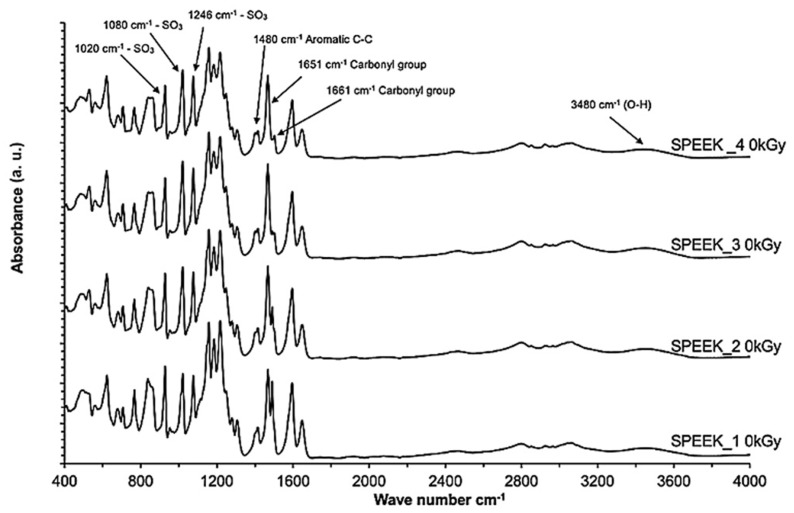
FT-IR spectra of non-irradiated SPEEK membranes with varying sulfonation degrees.

**Figure 7 materials-16-06098-f007:**
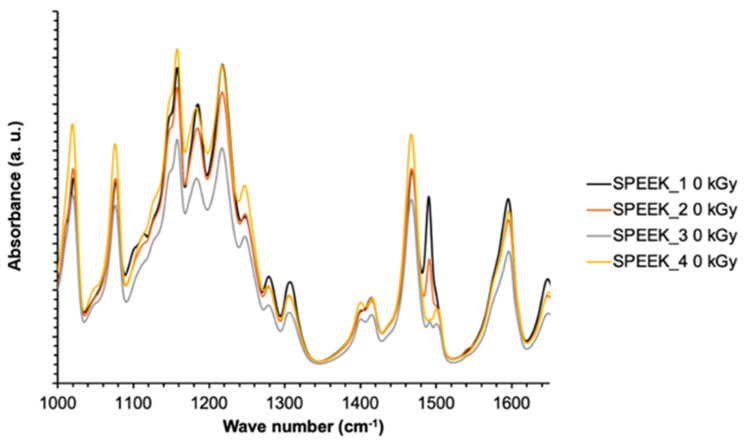
Normalized FT-IR spectra of non-irradiated SPEEK membranes.

**Figure 8 materials-16-06098-f008:**
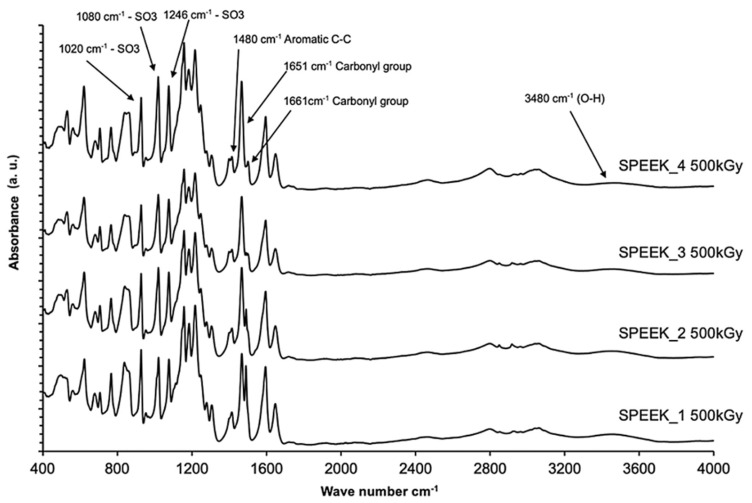
FT-IR spectra of irradiated SPEEK membranes with varying sulfonation degrees.

**Figure 9 materials-16-06098-f009:**
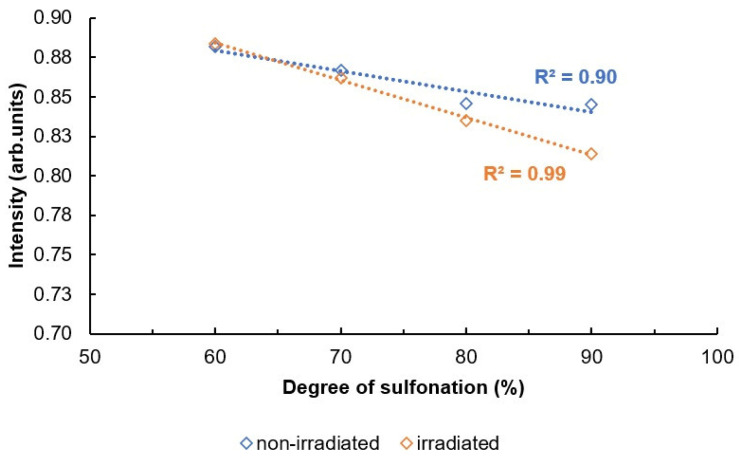
Intensities of the signals depending on the DS for 1185 cm^−1^.

**Figure 10 materials-16-06098-f010:**
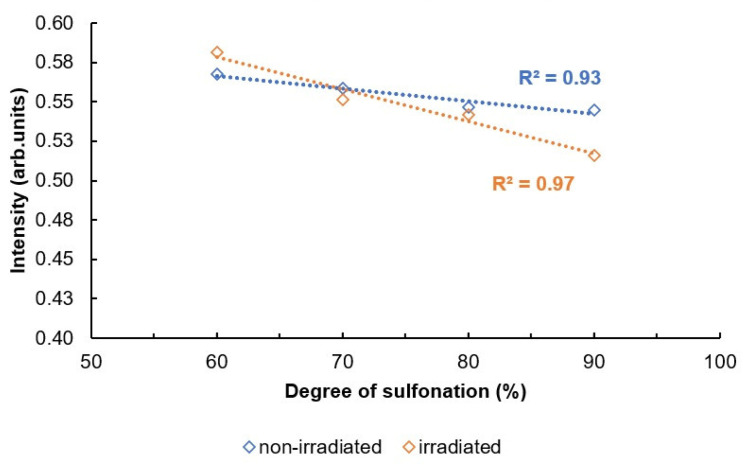
Intensities of the signals depending on the DS for 1596 cm^−1^.

**Figure 11 materials-16-06098-f011:**
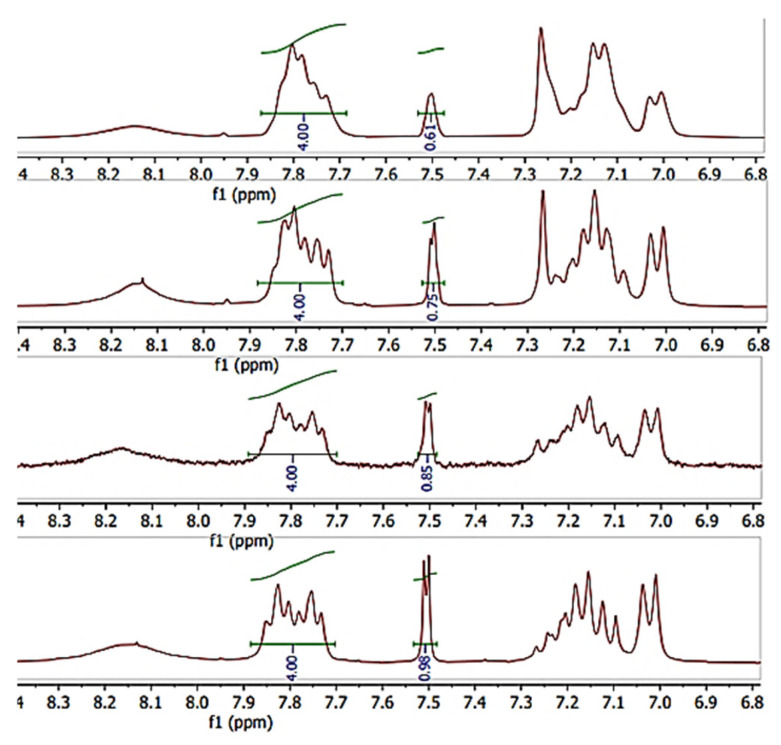
^1^H-NMR spectra of various DS (60–90%) non-irradiated SPEEK membranes.

**Figure 12 materials-16-06098-f012:**
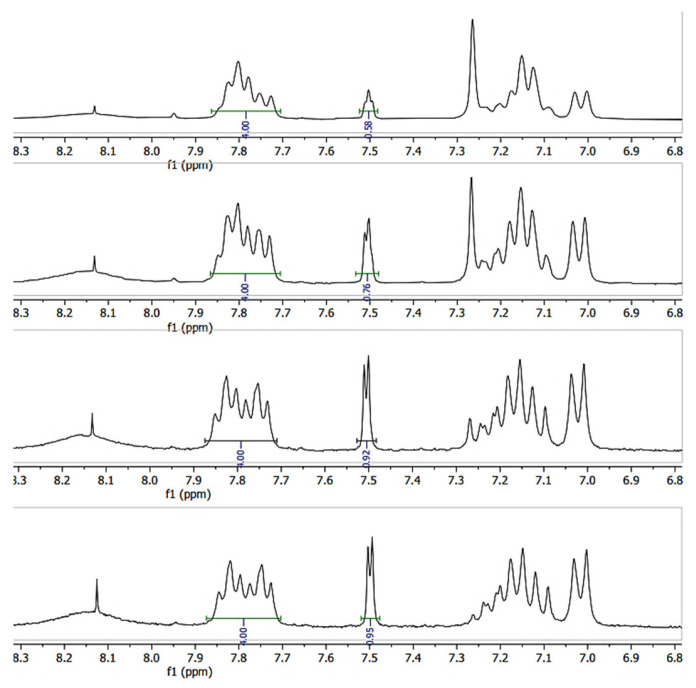
^1^H-NMR spectra of various DS (60–90%) irradiated SPEEK membranes.

**Figure 13 materials-16-06098-f013:**
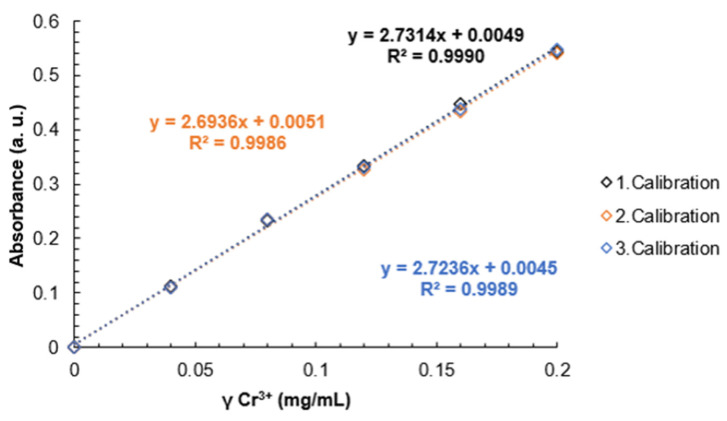
Calibration curves for Cr(III) standard solutions.

**Figure 14 materials-16-06098-f014:**
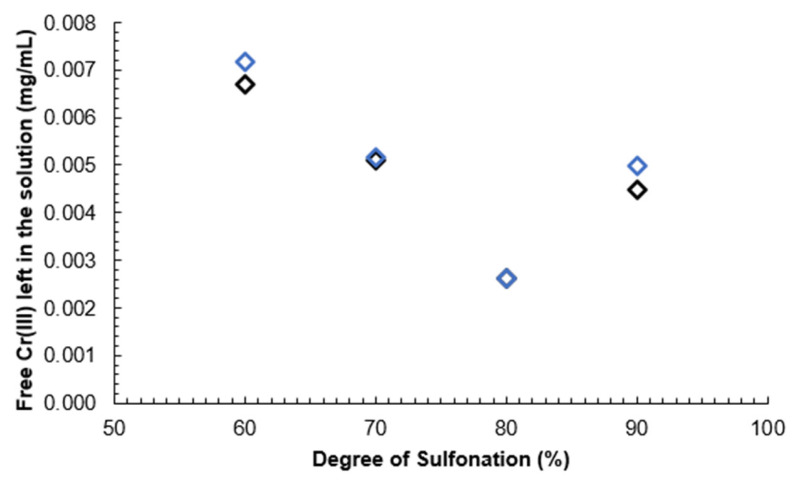
Unreacted Cr(III) mass concentration depending on the DS of the membrane.

**Figure 15 materials-16-06098-f015:**
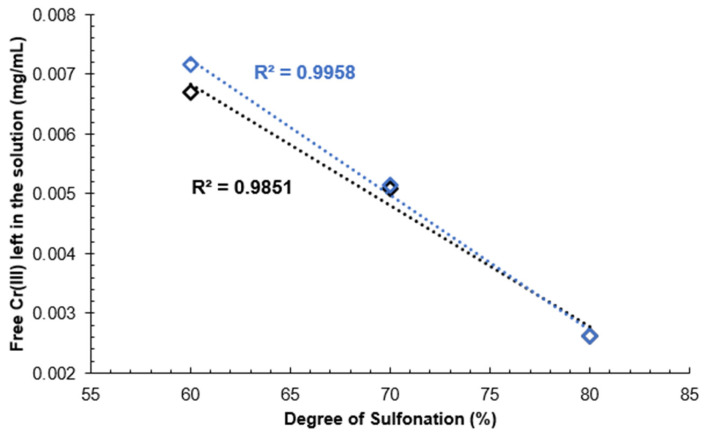
Linear range of unreacted Cr(III) mass concentration depending on the DS of the membrane.

**Figure 16 materials-16-06098-f016:**
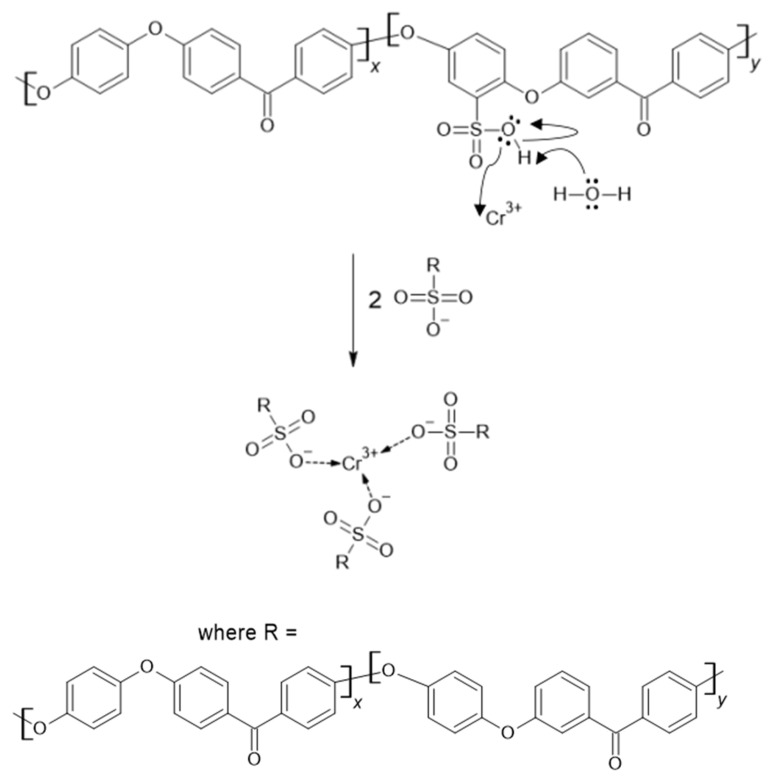
A theoretical mechanism for the binding of Cr(III) to SPEEK.

**Figure 17 materials-16-06098-f017:**
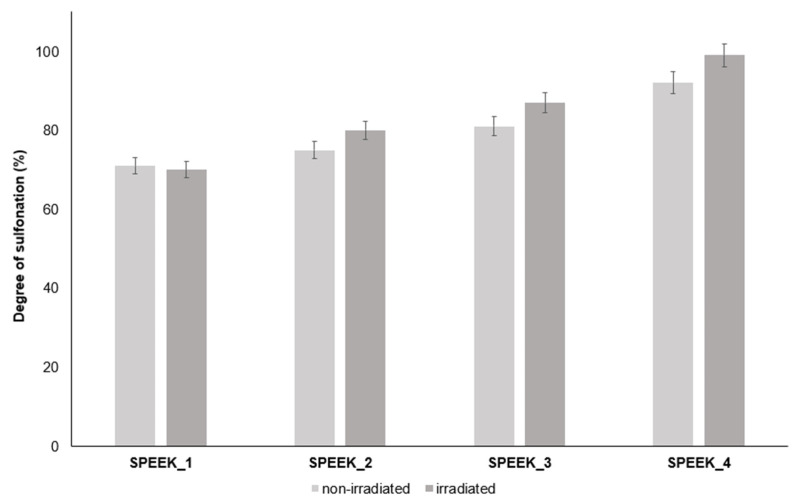
DS values obtained via TGA for non-irradiated and irradiated SPEEK membranes.

**Figure 18 materials-16-06098-f018:**
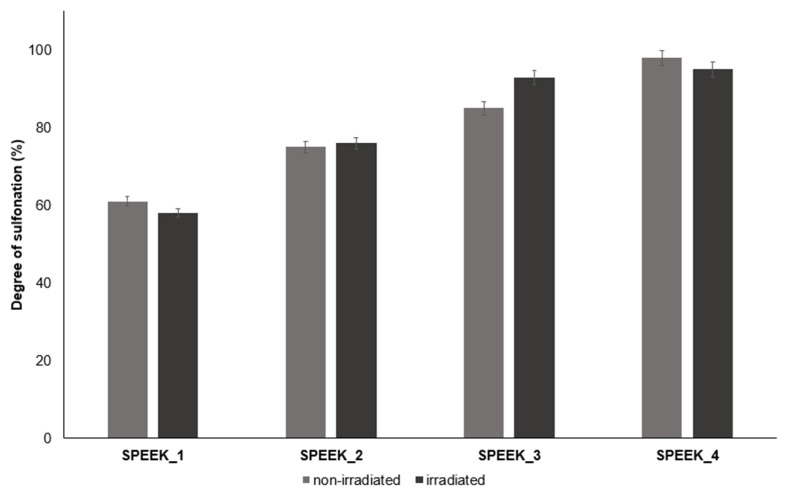
Obtained DS values by ^1^H-NMR for non-irradiated and irradiated SPEEK membranes.

**Table 1 materials-16-06098-t001:** SPEEK with different DS—synthesis parameters.

Sample No.	Duration, h	Temperature, °C
SPEEK_1	24	29
SPEEK_2	24	38
SPEEK_3	48	40
SPEEK_4	72	40

**Table 2 materials-16-06098-t002:** DS calculated via TGA.

Sample	SPEEK_1	SPEEK_2	SPEEK_3	SPEEK_4
Non-irradiated DS, %	71 ± 3	75 ± 3	81 ± 3	92 ± 3
Irradiated DS, %	70 ± 3	80 ± 3	87 ± 3	99 ± 3

**Table 3 materials-16-06098-t003:** DS calculated via ^1^H-NMR.

Sample	SPEEK_1	SPEEK_2	SPEEK_3	SPEEK_4
Non-irradiated DS, %	61 ± 2	75 ± 2	85 ± 2	98 ± 2
Irradiated DS, %	58 ± 2	76 ± 2	92 ± 2	95 ± 2

**Table 4 materials-16-06098-t004:** An overview of various methods and their advantages (+) and disadvantages (−) for the characterization of SPEEK DS.

	TGA	^1^H NMR	FT-IR	Spectrophotometry
Absolute value	+	+	−	−
Quick analysis time	−	+	+	+
Inexpensive equipment	−	−	−	+
No calibration curve required	+	+	−	−
Easily accessible	−	−	−	+
Non-degenerative sample analysis	−	−	+	−

## Data Availability

Data are contained within the article.

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
