# Peer review of "Critical Evaluation of the Methods for the Characterization of the Degree of Sulfonation for Electron Beam Irradiated and Non-Irradiated Sulfonated Poly(ether ether ketone) Membranes"

_materials, 2023, doi:10.3390/ma16186098_

Round 1

Reviewer 1 Report

The study investigated sulfonated poly(ether ether ketone) (SPEEK) materials as alternatives to Nafion™ in applications like fuel cells. SPEEK offers cost-effective, stable properties based on sulfonation level (DS). Various methods were used to determine DS in irradiated and non-irradiated membranes. Irradiated membranes showed higher DS, with changes seen in spectroscopy. A new Cr(III) method was effective up to 80% DS. This research informs about radiation effects on SPEEK. Some issues must be solved before it is considered for publication.

1.      This article shares lot of similarities with the published article (https://doi.org/10.1016/j.polymdegradstab.2022.109970) Authors should address the novelty compared to previously published one.

2.      What structural changes occur in SPEEK membranes when exposed to high-dose ionizing radiation, and how do these changes impact their performance? Can the newly proposed Cr(III) method effectively determine DS in sulfonated membranes with a DS of up to 80%?

3.      How do the different characterization techniques (FT-IR, TGA, 1H-NMR, electrochemical impedance, and Cr(III) spectrophotometry) compare in accurately assessing DS before and after irradiation? What are the underlying mechanisms responsible for the observed increase in DS in irradiated SPEEK membranes?

4.      How can the knowledge of radiation-induced effects on DS be utilized to enhance the design and performance of SPEEK membranes in fuel cell applications? Can the structural changes identified through spectroscopic analysis be correlated with changes in the thermal stability and proton conductivity of the SPEEK membranes?

5.      What are the potential limitations or challenges in scaling up the production of sulfonated poly(ether ether ketone) membranes with controlled DS for industrial applications? How do the performance and durability of irradiated and non-irradiated SPEEK membranes compare in real-world fuel cell operating conditions?

N/A

Author Response

Thank you for your comments. On behalf of my co-authors, we very much appreciate the time and effort you have put into your comments on our manuscript.

We have carefully reviewed the comments and thoroughly revised the manuscript accordingly. Our responses are given in a point-by-point manner below. We have submitted a revised version of our manuscript. All the changes are marked by using a red color for the text.

Reviewer 2 Report

The manuscript "Critical evaluation of the methods for the characterization of the degree of sulfonation for electron beam irradiated and non-irradiated sulfonated poly(ether ether ketone) membranes" aims at comparing different methods used for determination of the degree of sulfonation for sulfonated membranes before and after high dose irradiation.

The manuscript provides a detailed experimental study of sulfonated poly(ether ether ketone) (SPEEK) membranes with an appropriate set of experimental methods and the analysis results that seem to be convincing. The overall impression of the manuscript is that it can be considered for publication in the respective special issue of Materials provided that the following comments are addressed:

1) In the abstract, it is recommended to briefly outline the main findings of the article after stating its aim.

2) What was the thickness of the membranes? Was this parameter controlled during fabrication of the membranes? How could it affect the results of irradiation by an electron beam?

3) What are the errors of the data shown in Tables 2 and 3?

4) The authors should check the manuscript for punctuation typos such as that on Line 157 (Figure.3.) and so on.

5) Are the references formatted according to the journal template? Please check.

Author Response

Thank you for your comments. On behalf of my co-authors, we very much appreciate the time and effort you have put into your comments on our manuscript.

We have carefully reviewed the comments and thoroughly revised the manuscript accordingly. Our responses are given in a point-by-point manner below. We have submitted a revised version of our manuscript. All the changes are marked by a red text color.
